# The Protective Role of Caring Parenting Styles in Adolescent Bullying Victimization: The Effects of Family Function and Constructive Conflict Resolution

**DOI:** 10.3390/bs15070982

**Published:** 2025-07-19

**Authors:** Haoliang Zhu, Haojie Fu, Haiyan Liu, Bin Wang, Xiao Zhong

**Affiliations:** 1Department of Psychology, Wenzhou University, Wenzhou 325035, China; zhl1006@163.com; 2Shanghai Science Center for Autonomous Intelligent Unmanned Systems, Tongji University, Shanghai 201210, China; fuhaojie@tongji.edu.cn; 3Department of Military Medical Psychology, Air Force Medical University, Xi’an 710032, China; lhy0727ap@163.com; 4Department of Applied Psychology, Southwest University of Science and Technology, Mianyang 621010, China; 5School of Psychology, Beijing Sport University, Beijing 100084, China

**Keywords:** caring parenting style, bullying, family function, constructive conflict resolution

## Abstract

Based on attachment theory and the McMaster family functioning model, this study explores the protective role and mechanisms of a caring parenting style in protecting adolescents from bullying, from the perspective of the family environment. Study 1, conducted in Southwest China with middle school students (n = 4582), investigates the relationship between a caring parenting style and adolescent bullying victimization through a large-scale cross-sectional survey. The results show that both parents’ caring parenting styles are significantly negatively correlated with adolescent bullying victimization. Study 2, a two-wave study (n = 302), explores the protective mechanisms of a caring parenting style in adolescent bullying victimization. We not only observed again that a caring parenting style significantly negatively predicts bullying victimization but also found that family functioning and constructive conflict resolution play a chain-mediating role in this relationship. This finding not only supports the core hypothesis of attachment theory regarding the role of a secure base but also expands the theoretical model of bullying protection from a family ecological perspective by revealing a three-level transmission mechanism of parenting style–family system–individual capability, providing a theoretical anchor for the construction of a “family–school” collaborative intervention framework.

## 1. Introduction

Bullying refers to a form of aggressive behavior in which a more powerful individual repeatedly and intentionally inflicts physical or psychological harm on someone less powerful, resulting in physical or emotional distress ([28]). Bullying behaviors can be further categorized into traditional school bullying—such as physical aggression and verbal abuse—and cyberbullying, including malicious attacks on social media platforms ([8]). Globally, nearly one-third of students have reported being bullied by peers at school at least once in the past month ([23]). A large body of research has demonstrated that bullying victimization can significantly impair adolescents’ physical and mental development ([57]), with potential long-term consequences that may persist throughout their life ([25]; [50]). Given the profound impact of bullying on adolescent development, developing scientifically grounded and effective prevention and intervention strategies has become a critical focus in educational and psychological research.

### 1.1. Caring Parenting Styles and Bullying

The family is one of the primary environments children are exposed to during their development and plays a crucial role in their individual growth. Research has confirmed that family factors can influence the risk of children experiencing school bullying ([30]; [55]; [58]). According to the social learning theory, the environment in which an individual grows up has both a modeling and transmission effect ([44]). Interactions with parents during early life significantly influence children’s behavioral patterns ([21]). Parent–child relationships are not only the child’s first interpersonal relationships but also serve as a template for developing other social relationships, determining the quality of their peer friendships ([1]). Smooth communication between parents and children ([15]), reliable parental support and trust ([47]), and a positive family atmosphere ([16]) all serve as protective factors against peer victimization. Parenting style reflects and manifests the bidirectional interaction and its quality between the family system and adolescents ([4]), and it significantly impacts bullying. It can be broadly categorized into positive and negative approaches ([32]). Positive parenting emphasizes respect, support, and guidance, focusing on fostering healthy interactions between parents and children ([5]). It encompasses parenting practices characterized by caring, encouragement of autonomy, and appropriate control ([41]). Positive parenting styles help enhance children’s self-esteem ([38]) and effectively reduce the risk of both bullying and being bullied ([10]). In contrast, negative parenting styles are often characterized by neglect, excessive control, or inappropriate responsiveness, which may adversely affect children’s emotional development, social functioning, and self-concept ([5]). Such parenting practices have been associated with increased risks of both engaging in and becoming a target of bullying ([9]). Moreover, research has shown that adopting a positive and supportive parenting approach can facilitate the development of adolescents’ brain systems related to reward processing and emotion regulation, thereby mitigating bullying behaviors ([32]). However, the specific components of positive parenting that protect against bullying remain unclear.

Attachment theory posits that early attachment experiences with caregivers have long-term implications for individuals’ social relationships and stress regulation in adulthood ([48]). A caregiving style (caring parenting style) characterized by warmth, affection, closeness, and low psychological control may provide children with a “secure base” ([42]), which is essential for the development of secure attachment, interpersonal competence, and healthy personality functioning ([17]). Secure attachment is a positive state that alleviates threats ([11]). The extension–construction loop theory of secure attachment suggests that the activation of secure attachment significantly promotes positive self–other cognitions and virtues such as empathy and altruism ([37]; [53]), thus protecting adolescents who have experienced bullying. Based on this, the study proposes the following hypothesis:

**H1:** 
*A caring parenting style negatively predicts bullying victimization.*


### 1.2. The Role of Family Functioning and Constructive Conflict Resolution

Family functioning refers to the emotional connections and problem-solving abilities among family members within the family system ([35]). During children’s development, good family functioning provides valuable psychological resources and strength, helping individuals navigate potential crises, such as bullying ([36]). According to attachment theory, caring parenting styles provide adolescents with a warm, accepting, and supportive environment ([29]) and may also promote individual positive development through the integration of family functioning ([36]). Empirical studies show that strong family functioning can effectively counteract the bullying children encounter and reduce the risk of future bullying victimization ([51]). Furthermore, after experiencing bullying, warm parental support and a positive family atmosphere can enhance children’s physical and mental resilience, reducing the negative impacts of bullying on their development ([2]). Therefore, this study proposes the following hypothesis:

**H2:** 
*Family functioning mediates the relationship between a caring parenting style and bullying victimization, meaning that a caring parenting style reduces the likelihood of adolescent bullying through the enhancement of family functioning.*


Social dominance theory posits that individuals generally encourage unequal intergroup relations and support the dominance of certain groups over others. People exhibit varying degrees of social dominance orientation, which reflects the psychological tendency to support the dominance of “high-status” groups over “low-status” groups ([52]). Bullies often gain material resources and higher-level developmental needs, such as possessions, money, peer approval, and respect, through their bullying behavior ([45]). In response to bullying situations, there are three types of coping strategies: constructive conflict resolution, withdrawal, and seeking help from a third party ([7]; [39]). Constructive conflict resolution refers to efforts aimed at resolving disagreements, seeking consensus, and finding solutions ([7]). Individuals who adopt constructive conflict resolution strategies are typically active participants in problem-solving, using communication skills or other conflict management methods to address issues ([39]; [59]), thereby reducing bullying ([7]). Previous studies have also found a significant relationship between positive parenting styles and conflict resolution strategies ([26]; [31]).

The McMaster family functioning model suggests that the basic function of the family is to provide the necessary environmental conditions for the physiological, psychological, and social development of its members ([49]). Good family functioning, as an external protective factor for adolescent development, not only fosters a sense of trust and provides sufficient social support for individuals ([33]), but it also makes individuals more likely to adopt proactive coping strategies ([13]). Empirical studies have also shown that family functioning predicts children’s mental health and behavioral problems through coping strategies ([20]). Based on this, the study proposes the following research hypothesis:

**H3:** 
*Constructive conflict resolution mediates the relationship between a caring parenting style and bullying victimization, meaning that a caring parenting style reduces the likelihood of adolescent bullying through enhancing their constructive conflict resolution skills.*


**H4:** 
*Family functioning and constructive conflict resolution strategies play a chain mediation role in the relationship between a caring parenting style and bullying victimization. Specifically, a caring parenting style enhances family functioning, which in turn promotes adolescents’ constructive conflict resolution abilities, ultimately reducing the likelihood of adolescents being bullied.*


### 1.3. Current Study

Based on attachment theory and the McMaster Family Functioning Model, we conducted two studies to explore the protective role of a caring parenting style in adolescent bullying victimization and its underlying mechanisms. First, we employed a large-sample cross-sectional study to investigate whether both the father’s and mother’s caring parenting style offer protective effects against adolescent bullying. Then, we used a two-wave design (with a 6-month interval) to assess parenting styles, family functioning, conflict resolution strategies, and the degree of bullying. This longitudinal approach aimed to explore the potential mechanisms through which caring parenting styles protect adolescents from bullying over time, providing more direct guidance and support for the prevention and intervention of adolescent bullying behavior.

## 2. Study 1

### 2.1. Methods

#### 2.1.1. Participants

From May 2022 to December 2023, a total of 4889 first- and second-year high school students were recruited from seven high schools of the same level in Sichuan Province, China. The researchers conducted a group computer-based survey in the schools’ computer classrooms, collecting demographic data, as well as measuring the students’ parenting styles and bullying levels. Participants with low response accuracy were excluded using attention-check questions, leaving 4582 students for analysis. The study was approved by the school ethics committee. Informed consent was obtained from students’ parents, and both the school mental health coordinators and the participating students provided written informed consent prior to data collection.

#### 2.1.2. Measures

Parenting Style: The Parental Bonding Instrument (PBI) is a self-report questionnaire used to assess individuals’ perceptions of their parents’ parenting styles during childhood (before the age of 16). It consists of separate versions for mothers (PBI-M) and fathers (PBI-F), each with 23 items, and is divided into three subscales: Care, Encouragement of Autonomy, and Control ([41]). Previous studies have shown that the PBI is an effective and reliable measure of the parent–child relationship ([27]). In this study, the Care subscale was used to assess the caregiving parenting style of participants’ parents. The scale uses a 4-point Likert scale, where “0” means “strongly disagree,” “1” means “somewhat disagree,” “2” means “somewhat agree,” and “3” means “strongly agree.” Higher scores indicate a greater tendency toward a caring parenting style. The Cronbach’s α coefficient for the mother version of the scale was 0.799, and for the father version, it was 0.771 in this study.

School Bullying: The study used the California Bullying Victimization Scale to assess the extent of bullying experienced by participants ([18]). The scale asks respondents about the seven types of bullying victimization that they may have experienced at school, including verbal bullying (e.g., being called derogatory names, teased, or mocked), physical bullying (e.g., being hit, kicked, pushed, shoved, or locked in a room), and indirect bullying (e.g., being intentionally excluded from activities, left out of peer groups, or ignored). Students were asked to rate the frequency of each experience on a 5-point scale (0 = never, 1 = once in the last month, 2 = 2 or 3 times in the last month, 3 = about once a week, and 4 = several times a week). Higher scores indicate a higher level of bullying victimization. The Cronbach’s α coefficient for this scale in this study was 0.831.

#### 2.1.3. Data Analysis

In this study, quantitative data are presented as means and standard deviations, while qualitative data are presented as percentages. All data were analyzed using SPSS 25.0 and JASP 0.19.10. An independent samples t-test was conducted to examine gender differences in bullying, with effect sizes assessed using Cohen’s d (small = 0.2, medium = 0.5, and large = 0.8; [14]). Bayes factors (BF_10_) were also calculated to complement the *t*-test results ([54]). Pearson product–moment correlation analyses were performed to examine pairwise associations among the study variables.

### 2.2. Results

#### 2.2.1. Common Method Bias Test

Since this study primarily used a questionnaire survey to measure middle school students and all questionnaire data were collected and organized, the relationships between variables might be affected by common method bias ([34]). The scales used in this study had good reliability and validity, and Harman’s single-factor test was conducted to assess the impact of common method bias. The results showed that the first unrotated factor explained 28.533% of the total variance, which is below the 50% threshold ([34]), indicating that the results of this study were minimally affected by common method bias.

#### 2.2.2. Current Status of Adolescent Bullying Victimization

Basic information about the participants is presented in Table 1. Among the 4582 participants, there were 2278 male students and 2299 female students. The levels of verbal bullying, physical bullying, indirect bullying, and total bullying victimization were all significantly higher in boys compared to girls.

#### 2.2.3. Relationship Between Caring Parenting Style and Adolescent Bullying Victimization

A Pearson correlation analysis revealed a significant negative correlation between parental caring parenting style (average scores of both mother and father) and adolescent bullying victimization (*r* = −0.232, *p* < 0.001, and effect size = 0.236; Table 2). This indicates that the higher the level of a caring parenting style, the lower the degree of bullying victimization experienced by adolescents.

### 2.3. Discussion

To explore the relationship between a caring parenting style and adolescent bullying victimization, this study used a large sample cross-sectional design and found a significant negative predictive effect of a caring parenting style on the level of bullying victimization. In other words, the more parents adopt a caring parenting style, the less likely adolescents are to experience bullying. This result supports Hypothesis 1. According to attachment theory, a warm, understanding, and supportive parenting style helps foster secure attachment, which in turn promotes positive self–other cognitions, empathy, and altruistic behaviors ([37]; [53]). Previous studies have found that maternal psychological control (a negative parenting style) is more strongly associated with both bullying perpetration and victimization compared to paternal psychological control ([10]). Although we did not observe significant parental differences in the caring parenting style (positive), we found that the relationship between maternal care and bullying was closer than that between paternal care and bullying. This explanation indicates that there are also different positions of the parents in children’s development in positive parenting.

Additionally, we found that boys reported higher levels of bullying victimization than girls, including verbal, physical, and indirect bullying. This finding aligns with previous research ([7]; [10]). Socio-cultural expectations of rigid masculinity may create a dual mechanism ([6]; [43]): on one hand, it strengthens boys’ behaviors as potential perpetrators of bullying, and on the other hand, it suppresses their willingness to seek help as victims. This contradictory effect may increase the risk of victimization for male adolescents.

## 3. Study 2

### 3.1. Methods

#### 3.1.1. Participants

A separate sample of 356 first- and second-year high school students was recruited from another secondary school. Data collection was conducted in group sessions using computers in the school’s computer lab. At Time 1 (T1; May 2023), students completed measures of demographic variables, perceived parenting styles, family functioning, conflict coping strategies, and experiences of school bullying. Approximately six months later, at Time 2 (T2), family functioning, conflict coping strategies, and bullying experiences were reassessed. Informed consent was obtained from students’ parents, and both the school mental health coordinators and the participating students provided written informed consent prior to data collection. After excluding participants with missing data on key variables at either time point—including students with divorced or deceased parents—a total of 302 students completed both assessments and passed the attention-check items, yielding an overall attrition rate of 15.17%.

#### 3.1.2. Measures

Parenting Style: Same as in Study 1. In this study, Cronbach’s α coefficient for this scale was 0.719.

School Bullying: Same as in Study 1. In this study, Cronbach’s α coefficient for this scale was 0.858.

Conflict Resolution Style: This study used the Constructive Problem-Solving subscale from [40] ([40]) Conflict Resolution Questionnaire. The scale asks respondents about the coping strategies they might adopt when facing conflict. The responses are scored on a 5-point Likert scale: “0” = “Never”, “1” = “Occasionally”, “2” = “Sometimes”, “3” = “Often”, and “4” = “Always”. The higher the score, the more the individual tends to use constructive problem-solving. In this study, Cronbach’s α coefficient for this scale was 0.848.

Family Functioning: This study used the six positive items from the General Functioning subscale (GF6+) to assess family functioning ([3]). The scale uses a 4-point Likert scale: “1” means “Very much like my family”, “2” means “Like my family”, “3” means “Not like my family”, and “4” means “Completely unlike my family”. For calculation purposes, all items were reverse-coded, so higher scores on the scale indicate better family functioning. In this study, Cronbach’s α coefficient for this scale was 0.925.

#### 3.1.3. Data Analysis

In this study, quantitative data were presented as means and standard deviations, while qualitative data were presented as percentages. All data were analyzed using SPSS 25.0 and JASP 0.19.10. Pearson’s correlation analyses were conducted to examine pairwise associations among all variables (with point–biserial correlations used for the dichotomous gender variable). A chain mediation model (Model 6) from Hayes’s PROCESS macro for SPSS was then employed, with caring parenting style at Time 1 (T1) as the predictor variable and bullying victimization at Time 2 (T2) as the outcome variable. Family functioning (T2) and constructive conflict resolution strategies (T2) served as sequential mediators. Gender, as well as baseline levels of family functioning (T1), constructive conflict resolution (T1), and bullying victimization (T1), were included as covariates. The significance of indirect effects was tested using a bias-corrected percentile bootstrap estimation with 5000 resamples. Mediation effects were considered statistically significant if the 95% bootstrap confidence intervals did not include zero.

### 3.2. Results

#### 3.2.1. Common Method Bias Test

The results of the Harman single-factor test indicated that the first unrotated factor for T1 explained 25.985% of the total variance, and the first unrotated factor for T2 explained 39.492% of the total variance. Both percentages are below the critical value of 50% ([34]), suggesting that the results of this study were minimally affected by common method bias.

#### 3.2.2. Descriptive Statistics Results

Correlation analysis revealed that gender was significantly positively correlated with family functioning (T2). Parental bonding was significantly positively correlated with family functioning and constructive conflict resolution strategies (T1 and T2) and significantly negatively correlated with bullying victimization (T1 and T2). Family functioning (T2) was significantly positively correlated with constructive conflict resolution (T2) and significantly negatively correlated with bullying victimization (T2). Constructive conflict resolution strategies (T2) were significantly negatively correlated with bullying victimization (T2) (see Table 3).

#### 3.2.3. Chain Mediation Model Results

Based on the correlation analysis results, with gender, family functioning (T1), constructive conflict resolution strategies (T1), and bullying experiences (T1) as control variables, caring (T1) as the predictor variable, bullying (T2) as the outcome variable, and family functioning (T2) and constructive conflict resolution (T2) as the mediators, we conducted a chain mediation effect test using Model 6 of the Hayes’s SPSS PROCESS macro. The results are shown in Table 4. Caring positively predicted family functioning (T2) (B = 0.210, *p* = 0.006), supporting Hypothesis H1. Additionally, family functioning (T2) positively predicted constructive conflict resolution (T2) (B = 0.341, *p* < 0.001) and negatively predicted bullying experiences (T2) (B = −0.247, *p* < 0.001). Constructive conflict resolution (T2) negatively predicted bullying experiences (T2) (B = −0.078, *p* = 0.003).

The mediation effect was analyzed using the bootstrap mediation test. The results (Table 5 and Figure 1) show that the direct effect of caring (T1) on bullying experiences (T2) was not significant (effect value = −0.049, 95%CI = [−0.149, 0.051]). However, family functioning (T2) significantly mediated the relationship between caring (T1) and bullying experiences (T2) (effect value = −0.052, 95%CI = [−0.117, −0.007]), supporting Hypothesis H2. The mediation effect of constructive conflict resolution (T2) between caring (T1) and bullying experiences (T2) was not significant (effect value = −0.009, 95%CI = [−0.034, 0.008]), not supporting Hypothesis H3. However, the chain mediation effect of family functioning (T2) and constructive conflict resolution (T2) between caring (T1) and bullying experiences (T2) was significant (effect value = −0.006, 95%CI = [−0.019, −0.0002]), supporting Hypothesis H4.

### 3.3. Discussion

To explore the potential protective mechanism of a caring parenting style on adolescent bullying victimization, this study recruited 302 middle school students from two high schools in Sichuan Province, China, and further validated the protective effect of a caring parenting style on adolescent bullying. Based on attachment theory and the McMaster Family Functioning Theory, we also investigated the role of family functioning and constructive conflict resolution strategies in the relationship between a caring parenting style and bullying. The results revealed that a caring parenting style can alleviate bullying by enhancing family functioning and constructive conflict resolution abilities, thus confirming a three-level transmission mechanism involving parenting style, family systems, and individual abilities.

First, this study found that family functioning mediated the relationship between a caring parenting style and bullying victimization. This supports Hypothesis H2, which posits that a caring parenting style reduces the likelihood of adolescent bullying by enhancing family functioning. This finding is consistent with previous research ([51]). Attachment theory suggests that parental care—characterized by affection, gentleness, closeness, and low control—provides children with a “secure base” ([17]). A secure attachment relationship helps to promote good family functioning ([35]), providing individuals with ample psychological resources and strength to navigate potential developmental crises, such as bullying ([36]). This finding confirms the core hypothesis of attachment theory regarding the role of the secure base.

Furthermore, although we did not observe a significant individual mediating effect of constructive conflict resolution in the relationship between a caring parenting style and bullying victimization, we found that the chain mediation effect through family functioning and constructive conflict resolution was significant. This finding supports Hypothesis 4. The result suggests that a single caring parenting style may not directly enhance children’s constructive conflict resolution skills; rather, the development of such skills likely requires the joint effort of the entire family, specifically through establishing a family functioning system that provides an environment conducive to the physiological, psychological, and social well-being of its members ([49]). Previous research has shown that adolescents who frequently experience poor family functioning are more prone to negative emotions ([12]) and are less likely to employ effective problem-solving strategies ([56]), which further supports our findings. In conclusion, our findings reveal a three-level transmission mechanism of parenting style–family system–individual ability and further extend the theoretical model of bullying protection from a family ecological perspective.

## 4. General Discussion

This study systematically examines the protective effect of a caring parenting style on adolescent bullying victimization and its underlying mechanisms through a cross-sectional survey (n = 4582) and a 6-month follow-up study (n = 302). We found that both parents’ caring parenting styles significantly negatively predicted bullying victimization, with family functioning and constructive conflict resolution strategies playing a chain-mediating role in this relationship. This finding not only supports the core hypothesis of attachment theory regarding the role of a secure base but also extends the theoretical model of bullying prevention from a family ecological perspective by revealing a three-level transmission mechanism of parenting style, family system, and individual capabilities.

The results of this study emphasize the protective role of a caring parenting style in adolescents’ mental health and, by introducing family functioning as a key mediating variable, reveal how optimizing the family environment can reduce the likelihood of adolescents experiencing bullying by enhancing family functioning. This finding provides important theoretical support for family interventions and highlights the core role of the family in adolescents’ mental health ([22]). Parents should recognize the importance of warm, supportive, and understanding parenting for adolescents and work to create a loving family environment. Family intervention programs can focus on improving family functioning, helping parents enhance their parenting styles, and strengthening family cohesion and communication, thereby reducing the likelihood of bullying.

The study further found that improving constructive conflict resolution skills is a crucial pathway through which caring parenting reduces adolescent bullying victimization via family functioning. Schools and families should prioritize developing adolescents’ problem-solving skills by organizing skill-training activities (e.g., [46]) to help them master effective coping strategies ([24]). In summary, enhancing a caring parenting style, optimizing family functioning, and fostering adolescents’ constructive conflict resolution abilities can effectively reduce the likelihood of adolescents experiencing bullying.

This study has several limitations. First, the sample was limited to the southwest region of China, and data from a broader range of regions within China were not included. Therefore, the results may not fully represent the cultural context of China. Previous research has found that, compared to other countries and ethnic groups, psychological control and harsh parenting styles are more strongly associated with bullying in studies involving Chinese and Asian populations ([10]). Future research could collect data from participants with different cultural backgrounds and conduct cross-cultural studies. Second, this study tracked participants’ psychological status for only six months and focused solely on constructive conflict resolution as a coping strategy. Future research could extend the follow-up period and include a broader range of conflict coping strategies to yield more comprehensive findings. Third, due to some parents’ sensitivity regarding questions about family socioeconomic status, this study did not examine the impact of this important factor on the model. Future research could investigate whether socioeconomic status influences the model with guardian consent. Finally, future research should design intervention strategies related to the findings of this study and conduct family–school collaborative intervention studies ([19]). For example, intervention programs aimed at improving family functioning and cultivating constructive conflict resolution skills could be developed and evaluated for their effectiveness in preventing adolescent bullying behaviors. This would provide more direct guidance and support for practical applications.

## 5. Conclusions

This study integrates Bowlby’s attachment theory’s “secure base” mechanism with the systemic perspective of the McMaster Family Functioning Model. By adopting a mixed research design (cross-sectional survey and two-wave study), it reveals the protective mechanism of caring parenting styles against adolescent bullying victimization. Specifically, both parents’ caring parenting styles exert a lasting protective effect on bullying victimization through a chain-mediated pathway of “family functioning → constructive conflict resolution strategies.” This finding confirms the three-tiered transmission model in which caring parenting influences the development of individuals’ socialization skills by shaping family system operation patterns. The research results, from a developmental ecological perspective, provide a theoretical anchor for constructing a “family–school” collaborative intervention framework. It suggests that future anti-bullying practices should simultaneously enhance parenting quality, optimize family functioning effectiveness, and cultivate adaptive coping skills in adolescents.

## Figures and Tables

**Figure 1 behavsci-15-00982-f001:**
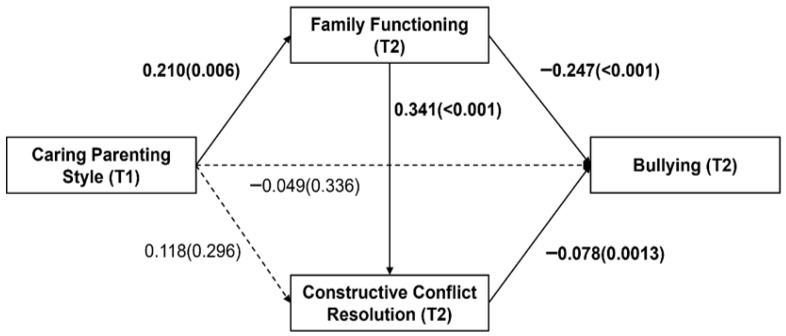
Chain mediation model. Solid lines represent *p* < 0.05, while dashed lines represent *p* > 0.05.

**Table 1 behavsci-15-00982-t001:** Basic information about participants and bullying victimization levels (N = 4582).

Variable	Male (n = 2278)	Female (n = 2299)	*t*	*p*	Cohen’s d	BF_10_
Age	15.107 ± 1.1763	15.010 ± 1.750	1.864	0.062	0.055	0.188
Verbal bullying	1.406 ± 0.789	1.308 ± 0.682	4.498	<0.001	0.133	781.405
Physical bullying	1.156 ± 0.525	1.070 ± 0.309	6.767	<0.001	0.200	2.437 × 10^+8^
Indirect bullying	1.324 ± 0.721	1.269 ± 0.561	2.857	0.004	0.084	1.937
Total bullying	1.286 ± 0.576	1.204 ± 0.412	5.504	<0.001	0.163	114,331.012

**Table 2 behavsci-15-00982-t002:** The results of the Pearson correlation analysis in study 1.

Variable	M ± SD	1	2	3	4
Bullying	1.245 ± 0.502	—			
Mother Caring	2.096 ± 0.665	−0.227 (<0.001)	—		
Father Caring	1.962 ± 0.712	−0.183 (<0.001)	0.550 (<0.001)	—	
Parental Caring	2.029 ± 0.607	−0.232 (<0.001)	0.871 (<0.001)	0.889 (<0.001)	—

**Table 3 behavsci-15-00982-t003:** The results of the Pearson correlation analysis in study 2.

Variable	1	2	3	4	5	6	7	8
Gender	1							
Caring T1	0.055 (0.344)	1						
Family Functioning T1	0.098 (0.089)	0.593 (<0.001)	1					
Constructive Conflict Resolution T1	0.103 (0.074)	0.357 (<0.001)	0.274 (<0.001)	1				
Bullying T1	0.075 (0.191)	−0.246 (<0.001)	−0.220 (<0.001)	−0.160 (0.005)	1			
Family Functioning T2	0.133 (0.021)	0.363 (<0.001)	0.405 (<0.001)	0.197 (0.001)	−0.031 (0.596)	1		
Constructive Conflict Resolution T2	0.096 (0.096)	0.283 (<0.001)	0.241 (<0.001)	0.402 (<0.001)	−0.121 (0.035)	0.323 (<0.001)	1	
Bullying T2	0.062 (0.281)	−0.245 (<0.001)	−0.223 (<0.001)	−0.096 (0.095)	0.186 (0.001)	−0.426 (<0.001)	−0.283 (<0.001)	1
M	—	2.009	3.348	2.708	1.344	3.309	2.754	1.208
SD	—	0.580	0.620	0.978	0.477	0.649	0.980	0.441

Note: Gender was coded as male = 1 and female = 0. T1 refers to Time Point 1 (baseline assessment), and T2 refers to Time Point 2 (follow-up assessment).

**Table 4 behavsci-15-00982-t004:** The results of the Regression analysis.

Outcome Variables	Predictor Variables	*B*	*SE*	*t*	*p*	95%CI	*R* ^2^	*F*
Family functioning T2	caring T1	0.210	0.075	2.796	0.006	[0.062, 0.357]	0.204	15.194 **
	gender	0.107	0.068	1.567	0.118	[−0.027, 0.241]		
	family functioning T1	0.302	0.068	4.433	<0.001	[0.168, 0.436]		
	constructive conflict resolution T1	0.037	0.037	0.994	0.321	[−0.036, 0.110]		
	bullying T1	0.111	0.074	1.506	0.133	[−0.034, 0.256]		
	constant	1.574	0.239	6.580	<0.001	[1.103, 2.045]		
Constructive conflict resolution T2	family functioning T2	0.341	0.086	3.955	<0.001	[0.172, 0.511]	0.232	14.829 **
	caring T1	0.118	0.112	1.047	0.296	[−0.104, 0.340]		
	gender	0.063	0.102	0.617	0.538	[−0.137, 0.263]		
	family functioning T1	0.011	0.105	0.101	0.920	[−0.195, 0.216]		
	constructive conflict resolution T1	0.320	0.055	5.782	<0.001	[0.211, 0.429]		
	bullying T1	−0.096	0.110	−0.875	0.382	[−0.313, 0.120]		
	constant	0.583	0.380	1.533	0.126	[−0.027, 0.241]		
Bullying T2	constructive conflict resolution T2	−0.078	0.026	−2.972	0.003	[−0.129, −0.026]	0.241	13.358 **
	family functioning T2	−0.247	0.040	−6.232	<0.001	[−0.326, −0.169]		
	caring T1	−0.049	0.051	−0.965	0.336	[−0.149, 0.051]		
	gender	0.098	0.046	2.154	0.032	[0.009, 0.188]		
	family functioning T1	−0.032	0.014	0.047	0.027	[−0.059, −0.004]		
	constructive conflict resolution T1	0.035	0.026	1.347	0.179	[−0.016, 0.087]		
	bullying T1	0.133	0.049	2.684	0.008	[0.035, 0.230]		
	constant	2.009	0.171	11.737	<0.001	[1.672, 2.345]		

*B*, unstandardized regression coefficient; *SE*, standard error. ** *p* < 0.001.

**Table 5 behavsci-15-00982-t005:** Bootstrap analysis of mediating effects.

	Effect	Boot *SE*	Boot LLCI	Boot UlCI
Total effect	−0.116	0.055	−0.223	−0.008
Direct effect	−0.049	0.051	−0.149	0.051
Total indirect effects	−0.067	0.038	−0.154	−0.010
Path 1 ^1^	−0.052	0.029	−0.117	−0.007
Path 2 ^2^	−0.009	0.011	−0.034	0.008
Path 3 ^3^	−0.006	0.005	−0.019	−0.0002

^1^ Path 1: Caring (T1) → Family Function (T2) → Bullying (T2); ^2^ Path 2: Caring (T1) → Constructive Conflict Resolution (T2) → Bullying (T2); and ^3^ Path 3: Caring (T1) → Family Function (T2) → Constructive Conflict Resolution (T2) → Bullying (T2).

## Data Availability

The data that support the findings of this study are available from the corresponding author, upon reasonable request.

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
