# Peer review of "The Protective Role of Caring Parenting Styles in Adolescent Bullying Victimization: The Effects of Family Function and Constructive Conflict Resolution"

_behavsci, 2025, doi:10.3390/bs15070982_

Round 1

Reviewer 1 Report

Comments and Suggestions for Authors

Dear author/s

Thank you for providing a very interesting paper to read. Your work is interesting and helpful. I would like to see some of the other bullying literature reviewed here, there is a long history that doesn't seem to be mentioned in this paper.

I'd also like to see a bit of a clearer link between the parenting theory and the bullying hypotheses. These are stated but a sentence here or there which steps this point out in each case would be very helpful to see your thinking and how you've posited those hypotheses.

Can you add a sentence that theorises why you don't think the data showed a direct mediating effect of parental atta chment (had to add a space so the system doesn't think I mean to add something to the comments) and other strategies on bullying experiences. I know you've said, "This result suggests that caring parenting style may not directly enhance a child's constructive conflict resolution abilities. Instead, it is necessary to establish a family functioning system that provides the environmental conditions for the healthy development of family members in terms of physiological, psychological, and social well-being" but why? Why do you think this finding is so? 

Limitations are very clear and rooted in the culture of the study participants.

Author Response

Dear Reviewer

We thank you for kindly reviewing our manuscript entitled " The Protective Role of Caring Parenting Styles in Adolescent Bullying Victimization: The Effects of Family Function and Constructive Conflict Resolution" (ID: behavsci-3678716). At the same time, thank you for your affirmation of our research. We agree with the comments. Enclosed please find both our point-by-point responses to the comments and the profoundly revised manuscript (modifications are highlighted in red). We appreciate you taking the time out of your busy schedule to review our revised manuscript and look forward to hearing from you soon.

Point-by-point responses:

Reviewer 1

R1-1: Thank you for providing a very interesting paper to read. Your work is interesting and helpful. I would like to see some of the other bullying literature reviewed here, there is a long history that doesn't seem to be mentioned in this paper.

A: Thanks for the comments. We were somewhat vague when introducing bullying. Therefore, under your suggestion, we have expanded the scope of bullying research and highlighted the attention of the current discipline to it. I think these changes will help readers understand the relevant research on bullying. We wrote that “Bullying refers to a form of aggressive behavior in which a more powerful individual repeatedly and intentionally inflicts physical or psychological harm on someone less powerful, resulting in physical or emotional distress (Juvonen & Graham, 2014). Bullying behaviors can be further categorized into traditional school bullying—such as physical aggression and verbal abuse—and cyberbullying, including malicious attacks on social media platforms (Camacho et al., 2023). Globally, nearly one-third of students have re-ported being bullied by peers at school at least once in the past month (Hong et al., 2022). A large body of research has demonstrated that bullying victimization can significantly impair adolescents’ physical and mental development (Yang et al., 2023), with potential long-term consequences that may persist throughout life (Hu, 2021; Takizawa et al., 2014). Given the profound impact of bullying on adolescent development, developing scientif-ically grounded and effective prevention and intervention strategies has become a critical focus in educational and psychological research.”(Lines 35-47)

R1-2: I'd also like to see a bit of a clearer link between the parenting theory and the bullying hypotheses. These are stated but a sentence here or there which steps this point out in each case would be very helpful to see your thinking and how you've posited those hypotheses.

A: Thanks for the comments. We were somewhat vague when describing the relationship between parenting and bullying. Following the suggestions from you and other reviewers, we revised this section, added concept definitions and theoretical explanations to make it clearer (Lines 49-88). We wrote that:

The family is one of the primary environment children are exposed to during their development and plays a crucial role in their individual growth. Research has confirmed that family factors can influence the risk of children experiencing school bullying (Khamis, 2015; Wang et al., 2021; Zhang et al., 2022). According to social learning theory, the environment in which an individual grows up has both a modeling and transmission effect (Price & Archbold, 1995). Interactions with parents during early life significantly influence children's behavioral patterns (Grusec, 2011). Parent-child relationships are not only the child's first interpersonal relationships but also serve as a template for developing other social relationships, determining the quality of their peer friendships (Boele et al., 2019). Smooth communication between parents and children (Doty et al., 2017), reliable parental support and trust (Sieving et al., 2017), and a positive family atmosphere (Es-pelage et al., 2018) all serve as protective factors against peer victimization. Parenting style reflects and manifests the bidirectional interaction and its quality between the fam-ily system and adolescents (Bowers et al., 2014), and it significantly impacts bullying. It can be broadly categorized into positive and negative approaches (Lee et al., 2024). Pos-itive parenting emphasizes respect, support, and guidance, focusing on fostering healthy interactions between parents and children (Bozoglan & Kumar, 2022). It encompasses parenting practices characterized by caring, encouragement of autonomy, and appropri-ate control (Parker, 1990). Positive parenting styles help enhance children's self-esteem (Milevsky et al., 2007) and effectively reduce the risk of both bullying and being bullied (Chu & Chen, 2024). In contrast, negative parenting styles are often characterized by ne-glect, excessive control, or inappropriate responsiveness, which may adversely affect children’s emotional development, social functioning, and self-concept (Bozoglan & Kumar, 2022). Such parenting practices have been associated with increased risks of both engaging in and becoming a target of bullying (Chen et al., 2022). Moreover, research has shown that adopting a positive and supportive parenting approach can facilitate the development of adolescents’ brain systems related to reward processing and emotion regulation, thereby mitigating bullying behaviors (Lee et al., 2024). However, the specific components of positive parenting that protect against bullying remain unclear.

Attachment theory posits that early attachment experiences with caregivers have long-term implications for individuals’ social relationships and stress regulation in adulthood (Slade & Holmes, 2019). A caregiving style characterized by warmth, affection, closeness, and low psychological control may provide children with a "secure base" (Rothbaum et al., 2002), which is essential for the development of secure attachment, in-terpersonal competence, and healthy personality functioning (Favaretto et al., 2001). Se-cure attachment is a positive state that alleviates threats (Colonnesi et al., 2011). The ex-tension-construction loop theory of secure attachment suggests that the activation of secure attachment significantly promotes positive self-other cognitions and virtues such as empathy and altruism (Mikulincer et al., 2011; Velotti et al., 2022), thus protecting adolescents who have experienced bullying.”

R1-3: Can you add a sentence that theorises why you don't think the data showed a direct mediating effect of parental attachment (had to add a space so the system doesn't think I mean to add something to the comments) and other strategies on bullying experiences. I know you've said, "This result suggests that caring parenting style may not directly enhance a child's constructive conflict resolution abilities. Instead, it is necessary to establish a family functioning system that provides the environmental conditions for the healthy development of family members in terms of physiological, psychological, and social well-being" but why? Why do you think this finding is so?

A: Thanks for the comments. In the introduction, we mentioned that “In response to bullying situations, there are three types of coping strategies: constructive conflict resolution, withdrawal, and seeking help from a third party (Butovskaya et al., 2007; Ohbuchi et al., 1996). Constructive conflict resolution refers to efforts aimed at re-solving disagreements, seeking consensus, and finding solutions (Butovskaya et al., 2007). Individuals who adopt constructive conflict resolution strategies are typically active participants in problem-solving, using communication skills or other conflict manage-ment methods to address issues (Ohbuchi et al., 1996; Zhao et al., 2015)”. Therefore, we hope to focus on individuals' conflict - coping styles, specifically the constructive conflict resolution strategy, from the perspective of positive psychology. This will help us develop corresponding social intervention programs to assist adolescents in actively participating in problem - solving when encountering conflicts, rather than withdrawing. It is possible that other strategies also have a certain impact on the experience of being bullied, but unfortunately, we have not paid attention to them yet. We have supplemented this deficiency in the limitations section (Lines 413-416).

At the same time, we also modified the explanation in the discussion according to your suggestion. We wrote that ”This finding supports Hypothesis 4. The result suggests that a single caring parenting style may not directly enhance children’s constructive conflict resolution skills; rather, the development of such skills likely requires the joint effort of the entire family, specifically through establishing a family functioning system that provides an environment condu-cive to the physiological, psychological, and social well-being of its members (Steven-son-Hinde & Akister, 1995).” (Lines 365-370)

Reviewer 2 Report

Comments and Suggestions for Authors

Thank you for the possibility to read this paper. The authors have developed a very interesting article composed of two studies analysing the sheltering role of protective parents and mechanisms of caring parenting style against bullying victimization. I have few suggestions that follow below:

ABSTRACT:

Pg 1. Line 22 “The results show that both parents' caring parenting styles are significantly negatively correlated with adolescent bullying victimization, con- firming the protective effect of nurturing parenting style.” – Authors are using the word nurturing with does not correspond correctly to “protective”. Please formulate

INTRODUCTION:

Pg. 2, Line 58 – Authors mention “positive parenting” and further is “negative parenting” – It is necessary to provide a definition what are the characteristics of positive or negative parenting. Rejection is the only concept identified for negative parenting, but it is unclear what do these concepts include.

Pg.2 Line 63 – “adopting appropriate parenting styles” – what are these parenting styles? Would be important previously explain the theory behind the parenting styles.

Pg. 2 Line 66 – “Attachment theory posits that parental care ….” – what is the reference for the attachment theory?

Pg.2. Line 75 – Hypothesis 1. what is caring parenting style? I believe the authors talk about parenting styles but don’t identify the caring parenting style. Is it positive parenting style, secure attachment?

STUDY 1.

METHODS:

When did the data collection of the first study occur? Did the authors ask the permission of students parents. Considering that the participants at less than 18 years old, they need authorization of parents to participate in studies.  It seems just the permission of the school and psychological health officers, but not the parents.

RESULTS:

Pg. 5 Line 195: Please provide a reference for Cohen d value and what are the ranges.

Moreover, it is important to explore also the magnitude of the correlations. The impact level is very low, therefore, the discussions should be made accordingly.

STUDY 2.

METHODS:

Pg. 6 Line 227 – Are the 302 participants part of the previous study or different students?

Moreover, in the Abstract, authors indicate that the study was a longitudinal study. This does not really apply to a longitudinal study, but a two-wave study. Practically the instruments were administered twice.

What were the criteria of inclusion and exclusion?  What about students that had two mothers or two fathers? What about students who did not have one of the parents, where they included?

What is the time frame of this study. Does that coincide with the first one?

RESULTS:

How was the Pearson test used with gender: How would you analyse gender and Caring parental style is positive? Which gender then is better? Nominal variables are not used for this test, because male and female participants is just a characteristic and does not mean that one is better than the other.

Figure 1. Family functioning, conflict resolutions and bullying are considered as T2, which makes it very difficult to follow the description of the results. What does T stand for?

CONCLUSION:

Pg. 11 Line 404. Please amend the longitudinal study. The 2nd study does not fulfil the criterion of a longitudinal study. It is only a two-wave study.  

Author Response

Dear Reviewer

We thank you for kindly reviewing our manuscript entitled " The Protective Role of Caring Parenting Styles in Adolescent Bullying Victimization: The Effects of Family Function and Constructive Conflict Resolution" (ID: behavsci-3678716). At the same time, thank you for your affirmation of our research. We agree with the comments. Enclosed please find both our point-by-point responses to the comments and the profoundly revised manuscript (modifications are highlighted in red). We appreciate you taking the time out of your busy schedule to review our revised manuscript and look forward to hearing from you soon.

Point-by-point responses:

ABSTRACT:

R2-1: Pg 1. Line 22 “The results show that both parents' caring parenting styles are significantly negatively correlated with adolescent bullying victimization, con- firming the protective effect of nurturing parenting style.” – Authors are using the word nurturing with does not correspond correctly to “protective”. Please formulate

A: Thanks for your careful reading. This is an ambiguity, and we have modified the way this sentence is described. We wrote that “The results show that both parents' caring parenting styles are significantly negatively correlated with adolescent bullying victimization.” (Lines 20-21)

INTRODUCTION:

R2-2: Pg. 2, Line 58 – Authors mention “positive parenting” and further is “negative parenting” – It is necessary to provide a definition what are the characteristics of positive or negative parenting. Rejection is the only concept identified for negative parenting, but it is unclear what do these concepts include.

A: Thanks for the comments. We didn't describe the concepts of relevant nouns during the writing process, which made this part of our work ambiguous. Therefore, under your suggestion, we added descriptions of the relevant concepts. This change has increased our readability. For example, “Positive parenting emphasizes respect, support, and guidance, focusing on fostering healthy interactions between parents and children (Bozoglan & Kumar, 2022). It encom-passes parenting practices characterized by caring, encouragement of autonomy, and appropriate control (Parker, 1990).” (Lines 63-66) And “in contrast, negative parenting styles are often characterized by neglect, excessive control, or inappropriate responsiveness, which may adversely affect children’s emotional de-velopment, social functioning, and self-concept (Bozoglan & Kumar, 2022). Such par-enting practices have been associated with increased risks of both engaging in and be-coming a target of bullying (Chen et al., 2022).” (Lines 68-72)

R2-3: Pg.2 Line 63 – “adopting appropriate parenting styles” – what are these parenting styles? Would be important previously explain the theory behind the parenting styles.

A: Thanks for the comments. This sentence was not clearly expressed, so we added references and proposed that adopting a positive and supportive parenting style is beneficial to the development of adolescents. We wrote that “Moreover, research has shown that adopting a positive and supportive parenting ap-proach can facilitate the development of adolescents’ brain systems related to reward processing and emotion regulation, thereby mitigating bullying behaviors (Lee et al., 2024).”( Lines 73-76)

R2-4: Pg. 2 Line 66 – “Attachment theory posits that parental care ….” – what is the reference for the attachment theory?

A: Thanks for the comments. We have reorganized the content of attachment theory according to your suggestions to make it clearer. We wrote that “Attachment theory posits that early attachment experiences with caregivers have long-term implications for individuals’ social relationships and stress regulation in adulthood (Slade & Holmes, 2019). A caregiving style characterized by warmth, affection, closeness, and low psychological control may provide children with a "secure base" (Rothbaum et al., 2002), which is essential for the development of secure attachment, in-terpersonal competence, and healthy personality functioning (Favaretto et al., 2001).”

R2-5: Pg.2. Line 75 – Hypothesis 1. what is caring parenting style? I believe the authors talk about parenting styles but don’t identify the caring parenting style. Is it positive parenting style, secure attachment?

A: Thanks for the comments. You have raised an important question. We didn't explain the caring parenting style. Therefore, when describing positive parenting styles, we stated that the caring parenting style is a positive way of parenting. At the same time, we also added its definition. We wrote that “A caregiving style (caring parenting style) characterized by warmth, affection, closeness, and low psychological control may provide children with a "secure base" (Rothbaum et al., 2002), which is essential for the development of secure attachment, interpersonal com-petence, and healthy personality functioning (Favaretto et al., 2001).” (Lines 80-83)

STUDY 1.

METHODS:

R2-6: When did the data collection of the first study occur? Did the authors ask the permission of students parents. Considering that the participants at less than 18 years old, they need authorization of parents to participate in studies.  It seems just the permission of the school and psychological health officers, but not the parents.

A: Thanks for the comments. We have added relevant details to make the research more detailed and comprehensive. We wrote that “From May 2022 to December 2023, a total of 4,889 first- and second-year high school students were recruited from seven high schools of the same level in Sichuan Province, China. The researchers conducted a group computer-based survey in the schools’ computer classrooms, collecting demographic data, as well as measuring the students’ parenting styles and bullying levels. Participants with low response accuracy were excluded using attention-check questions, leaving 4,582 students for analysis. The study was ap-proved by the school ethics committee. Informed consent was obtained from students’ parents, and both the school mental health coordinators and the participating students provided written informed consent prior to data collection.”

RESULTS:

R2-7: Pg. 5 Line 195: Please provide a reference for Cohen d value and what are the ranges.

A: Thanks for your careful reading. We added an introduction to the Cohen d value in the methods section. We wrote that “An independent samples t-test was conducted to examine gender differences in bullying, with effect sizes assessed using Cohen’s d (small = 0.2, medium = 0.5, large = 0.8; Cumming, 2014).” (Lines 193-195)

R2-8: Moreover, it is important to explore also the magnitude of the correlations. The impact level is very low, therefore, the discussions should be made accordingly.

A: Thanks for the comments. Our correlation analysis found that the relationship between maternal care and bullying is closer than that between paternal care and bullying. This is consistent with previous studies that have found that the relationship between mothers' psychological control (a negative parenting style) and bullying perpetration and victimization is more relevant than that of fathers' psychological control (Chu & Chen, 2024). From the perspective of positive parenting styles, we supplement the parental differences in childrearing. We wrote that “Previous studies have found that maternal psychological control (a negative parenting style) is more strongly associated with both bullying perpetration and victimization compared to paternal psychological control (Chu & Chen, 2024). Although we did not observe significant parental differences in the caring parenting style (positive), we found that the relationship between maternal care and bullying was closer than that between paternal care and bullying. This explanation indicates that there are also different posi-tions of parents in children's development in positive parenting.” (Lines 233-237)

STUDY 2.

METHODS:

R2-9: Pg. 6 Line 227 – Are the 302 participants part of the previous study or different students?

A: Thanks for your careful reading. These are additional students and we have added details. “A separate sample of 356 first- and second-year high school students was recruited from another secondary school.” (Line 248)

R2-10: Moreover, in the Abstract, authors indicate that the study was a longitudinal study. This does not really apply to a longitudinal study, but a two-wave study. Practically the instruments were administered twice.

A: Thank you for your reminder. We also agree with your opinion, so we have modified the expressions in the abstract and conclusion sections. (Lines 22 and 432).

R2-11: What were the criteria of inclusion and exclusion?  What about students that had two mothers or two fathers? What about students who did not have one of the parents, where they included?

A: Thank you for your reminder. You have pointed out an important detail. To avoid the influence of factors such as parental divorce and death, we have removed this part (21 people) during the initial data screening. We have added this part in the text. We wrote that “After excluding participants with missing data on key variables at either time point—including students with divorced or deceased parents—a total of 302 students completed both assessments and passed the attention-check items, yielding an overall attrition rate of 15.17%.” (Lines 256-259)

R2-12: What is the time frame of this study. Does that coincide with the first one?

A: Thank you for your reminder. We have added the measurement time information. We wrote that “Data collection was conducted in group sessions using computers in the school’s com-puter lab. At Time 1 (T1; May 2023), students completed measures of demographic var-iables, perceived parenting styles, family functioning, conflict coping strategies, and ex-periences of school bullying. Approximately six months later, at Time 2 (T2), family functioning, conflict coping strategies, and bullying experiences were reassessed.” (Lines 249-254)

RESULTS:

R2-13: How was the Pearson test used with gender: How would you analyse gender and Caring parental style is positive? Which gender then is better? Nominal variables are not used for this test, because male and female participants is just a characteristic and does not mean that one is better than the other.

A: Thanks for the comments. The correlation analysis between gender (binary classification) and continuous variables requires the use of point-biserial correlation. In SPSS, the operation of point-biserial correlation is exactly the same as that of Pearson, except that the binary classification variable needs to be scored as 0 and 1. To avoid ambiguity, "Pearson" in "Pearson correlation analysis shows" was deleted, and "correlation analysis shows" was retained. (Line 301)

R2-14: Figure 1. Family functioning, conflict resolutions and bullying are considered as T2, which makes it very difficult to follow the description of the results. What does T stand for?

A: T represents the test time. We have added details to the table's notes. We wrote that “Gender was coded as male = 1 and female = 0. T1 refers to Time Point 1 (baseline assessment), and T2 refers to Time Point 2 (follow-up assessment).” (Lines 310-311)

CONCLUSION: 

Pg. 11 Line 404. Please amend the longitudinal study. The 2nd study does not fulfil the criterion of a longitudinal study. It is only a two-wave study

A: Thanks for the comments. We changed the longitudinal study to a two-wave study.(Line 432)